# Risk Factors Associated with Multimorbidity among Children Aged Under-Five Years in Sub-Saharan African Countries: A Scoping Review

**DOI:** 10.3390/ijerph20021377

**Published:** 2023-01-12

**Authors:** Phillips Edomwonyi Obasohan, Stephen J. Walters, Richard Jacques, Khaled Khatab

**Affiliations:** 1School of Health and Related Research (ScHARR), University of Sheffield, Sheffield S1 4AD, UK; 2Department of Liberal Studies, College of Business and Administrative Studies, Niger State Polytechnic, Bida Campus, Bida 912231, Nigeria; 3Faculty of Health and Wellbeing, Sheffield Hallam University, Sheffield S10 2BP, UK

**Keywords:** multimorbidity, scoping review, under-five years, sub-Saharan Africa

## Abstract

Background/purpose: Globally, the prevalence of multimorbidity (defined as the cooccurrence of two or more diseases in an individual without reference to an index disease) is greater than 33%. Consequently, childhood multimorbidity, a growing public health concern in Low- and Middle-Income-Countries (LMICs), frequently has an impact on children’s health. Therefore, the aim of this scoping review was to locate and describe studies that investigate the association between socioeconomic, demographic, and environmental factors and the prevalence of multimorbidity among children aged under five years in Sub-Saharan Africa (SSA). Methods/Design: We searched MEDLINE, Cumulative Index to Nursing and Allied Health Literature (CINAHL), PubMed, Scopus, and Web of Science for papers written in English, and published between January 1990 and March 2022. The search included papers that focused on children aged under five years with multimorbidity from Sub-Saharan Africa and used classical regression methods in their analysis. Results: A total of 261 articles were identified. Out of the 66 articles selected for full-text reading, 60 were removed for various reasons. Therefore, data from a sample of six articles were finally extracted and reported in this study. The sample size for the 6 studies included ranged from 2343 to 193,065 children under five years of age. There were six distinct disease conditions (Pneumonia, diarrhoea, malaria, being overweight, stunting, and anaemia) analysed in the included studies. One of the studies had three concurrent diseases, while the other five studies had two current diseases as their multimorbidity outcome of interest. The prevalence of multimorbidity in these six studies ranged from 1.2% to 24.8%. Conclusions: The relatively few studies found in this research area is an indication of an evidence deficit/knowledge gap yearning to be filled to help policymakers in coming up with integrated multimorbidity care for children in SSA.

## 1. Introduction

Children’s health is often affected by childhood multimorbidity, a rising public health issue in developing nations. We define multimorbidity as the co-occurrence of two or more diseases in an individual without reference to an index disease. The frequency of multimorbidity is greater than 33% worldwide [1]. However, multimorbidity affects more than 70% of those aged 65 and older, making the elderly group the most at risk [2,3]. Children are not excluded, however, as research shows that a sizable proportion of children experience multimorbidity [4]. Numerous studies have linked multimorbidity to psychosocial factors, individual and social lifestyle choices (such as being overweight and obese, sedentary behaviour, smoking, and excessive alcohol consumption), and demographic changes (such as the ageing population and gender inequalities) [5]. The intricate way these disorders interact with socioeconomic, demographic, and environmental patterns presents the biggest challenge to clinicians, researchers, and healthcare professionals [2,3,6,7]. The most significant issues facing the healthcare system include the rise of multimorbidity in developing nations with subpar healthcare systems and a burden of chronic diseases. Therefore, it is important to carry out a scoping review of the risk factors associated with multimorbidity among children aged under five years in sub-Saharan African countries to contribute to the evidence-based healthcare decisions about the prevention of multimorbidity in SSA. To the best of our knowledge, previous reviews investigating the risk factors that influence the prevalence of multimorbidity in children under five years in SSA have not been published. Consequently, this research tries to close this gap.

### The Aim of the Scoping Review

The aim of this scoping review was to find and describe papers that described the prevalence of multimorbidity and examined the relationship between socioeconomic, demographic, and contextual characteristics and multimorbidity using classical statistical regression analysis methods among children aged under five years in Sub-Saharan Africa countries.

## 2. Materials and Methods

### 2.1. Design

This scoping review was tailored along with the enhanced framework of Arkey and O’Malley [8], the recommendations of Lecac et al. [9], and guidelines of the Agency for Healthcare Research and Quality (AHRQ) [10]. The five key steps include: (1) research question identification, (2) identification of the relevant study sources, (3) evidence and eligibility criteria selection, (4) data charting, and (5) results collating, summarising, and reporting [11]. However, the pattern of reporting the results in this scoping review followed the Preferred Reporting Items for Systematic Reviews and Meta-Analyses extension for Scoping Reviews (PRISMA-ScR) guidelines [12,13].

### 2.2. Identification of the Research Questions

We based the research question on the research purpose that was stated using the Population, Intervention, Comparators, Outcomes, Timing and Study design (PICOTS) framework of AHRQ [10]. The primary research question for this scoping review is: “what are the risk factors associated with multimorbidity among children under five years in sub-Saharan Africa countries”. Furthermore, the paper aims to answer the following questions:(i)What types of statistical analyses are employed in the existing evidence to determine the risk factors associated with multimorbidity among children under five?(ii)What are the various childhood disease combinations that formed the multimorbidity structures examined in the eligible studies?(iii)What are the multiple overlaps in the risk factors across the selected studies?(iv)What are the possible gaps in knowledge identified from the selected literature?

### 2.3. Eligibility Criteria

Studies included in the review followed the PICOTS criteria enumerated and defined as follows:

#### 2.3.1. Inclusion Criteria

(i)Population (P): The studies included male and female children under five years of age who resided in any Sub-Saharan Africa (SSA) country. The review also includes studies involving adults (or/and above five years children) and under five years children, provided data for under five were reported separately from any other age-groups.(ii)Intervention (I): Studies that focused on predictors or risk factors or determinants of multimorbidity among under five or preschool children in SSA that covered both individual and contextual exposures using classical frequentist statistical regression methods only.(iii)Comparator (C): The presence of two or more diseases versus no diseases was the focus of the comparators in the studies.(iv)Outcomes (O): Studies that involved two or more childhood diseases that were evaluated jointly rather than independently such that the outcomes reflect the interactions of the diseases.(v)Timing (T): The publication period for the article is between 1 January 1990 and 19 March 2022, to capture recent publications.(vi)Settings/Design (S): Observational studies, such as cross-sectional and longitudinal studies that focused on risk factors as exposures.

#### 2.3.2. Exclusion Criteria

(i)Studies that involved older children, but no separate data involving under five years were reported.(ii)Studies that do not meet the definition of ‘multimorbidity’ as the “cooccurrence of two or diseases among children under five years without reference to an index disease”.(iii)Studies not written in the English language.

#### 2.3.3. Steps Involved in the Inclusion and Exclusion Processes

The following priority steps were adopted in deciding whether to include or exclude a study report:If the study is multimorbidity, thenAre children aged under five years the unit of analysis? thenIs the country of analysis from SSA? thenDoes the study utilise a nationally representative survey in data collection?Does the study utilise classical frequentist statistical regression methods in the analysis?

If the answers to all five of these steps was ‘yes’, then the paper was included in the scoping review; otherwise, it was excluded from the review.

### 2.4. Identify the Relevant Sources of Evidence

#### 2.4.1. Information Sources

The literature search was conducted from MEDLINE, Cumulative Index to Nursing and Allied Health Literature (CINAHL), PubMed, Scopus, and Web of Science. It included only papers written in English, and the publication date was between January 1990 and March 2022. The final search was carried out on the 19 March 2022.

#### 2.4.2. Search Strategy

In this scoping review, the search terms were first entered into MEDLINE (Ovid), as shown in Table 1, by mapping terms to subject headings and limiting terms. Next, the search terms were derived from the PICOTS categories. These terms were then repeated for other databases consulted.

#### 2.4.3. Selection Process

PEO screened all the selected literature for titles and abstracts using the inclusion and extraction criteria as a benchmark. This process was done twice from two citation managers platforms (Endnote and Zotero). First on the 19 September 2021, and was repeated on the 19 March 2022, with the available papers updated. The full-text report was conducted for all the selected articles. Papers excluded were noted with reasons. This process was vetted independently by SJW, RJ, and KK.

#### 2.4.4. Data Charting Management

The data extracted from the included articles were first deposited into a Microsoft Excel sheet designed by the reviewer for this review. The relevant information extracted includes authors/year of publication, the paper title, study objectives, outcome of interest, the sample size (and age of participants), method of data analysis used, the procedure adopted (study design), and study location.

## 3. Results

The results section reports the profile of the quantitative analysis of risk factors associated with multimorbidity among children aged under five years in SSA, following the Preferred Reporting Items for Systematic Reviews and Meta-Analyses extension for Scoping Reviews (PRISMA-ScR) checklists [12,13].

### 3.1. Selection of Sources of Evidence

A total of 261 articles were identified from all the electronic databases consulted (MEDLINE(Ovid) = 12, CINAHL = 43, PUBMED= 124, Scopus = 27, Web of Science (WOS) = 50, other sources (from references) = 5), out of which 22 duplicates were removed (see Figure 1). Further, 173 were removed after the abstracts and titles were read for eligibility. Out of the 66 articles selected for full-text reading, an additional 60 were removed for various reasons (including (i) participants are above five years (6), (ii) not multimorbidity (MM) paper as per the definition of MM (26), (iii) Bayesian approach (4), (iv) non-national coverage (6), (v) Systematic reviews (3), and (vi) No full text found (15)). Therefore, data from a sample of six articles were finally extracted and reported in this study.

### 3.2. Characteristics of Sources of Evidence

To answer the scoping review questions raised previously, some relevant information extracted from the selected papers is contained in Table 2, Table 3, Table 4 and Table 5. The following section describes the characteristics of the sources of evidence.

### 3.3. Study Characteristics

In Table 2, all the papers were related to secondary analysis of nationally representative surveys. Five of the papers had a singular national setting, while one study was a multi-country study [19]. The sample size for the studies included range from 2343 to 193,065 children of under five years of age. There were six distinct disease conditions (Pneumonia, diarrhoea, malaria, overweight, stunting, and anaemia) analysed in the included studies. One of the studies [14] had three concurrent diseases, while the other five studies had two current diseases as their multimorbidity outcome of interest. The prevalence of the multimorbidity in these six studies ranged from 1.2% to 24.8%. Two of the studies applied multivariate logistic regression analysis [16,17] and one study each used generalized ordinal logistic regression analysis [14], multivariate poison regression model [15], mixed effect logistic regression [18], or multinomial logistic regression model [19]. Five of the studies included in the review used the Demographic and Health Survey (DHS) data set of their respective country of focus, while one used data from the multiple indicator cluster survey [15]. The survey years range from 2011 to 2016. Yet, the only multi-country study collected data from difference countries for surveys over a ten-year period from 2005 to 2015 [19].

#### Distribution of the Extracted Risk Factors of Multimorbidity

Table 3 presents the results of the statistical significance of each variable, classified into child-, parental-, household-, and community-related predictors, which were grouped into harmful or protective effects.

### 3.4. Child-Related Characteristics

#### 3.4.1. Child’s Age

The six studies included in the review reported significant association of child’s age with multimorbidity. Four papers (67%) [16,17,18,19] found that the prevalence of multimorbidity increased with the child’s age, with older children significantly more likely to be multimorbid compared to younger children. Duah et al. [16] found the opposite direction of effect, with increasing age associated with a reduced prevalence of multimorbidity, with the odds of having comorbid anaemia and diarrhoea for children aged 6–23 months twice that of the odds of children aged above 24 months, while Mulatya & Mutuku and Geda et al. found a higher odds of having contracted comorbid diarrhoea and pneumonia (ARI) and concurrent stunting and anaemia, respectively, for older children compared to children aged under 1 year [17,18]. On the contrary, Adedokun [14] found more than twice the protective effects for older children aged three years and above contracting multimorbidity of pneumonia, diarrhoea, and malaria versus combined none of the diseases, one of the diseases, and two of the diseases when compared with children aged less than one year. Similarly, Atsu et al. [15] found a borderline significant protective effect for children aged 12–23 months compared to children aged less than 12 months, and Tran et al. [19] found a near twice protective (risk) effect for being healthy versus comorbidity of stunting and anaemia in children aged 1–2 years compared to children aged 0.5–1 year. In other words, children aged 1–2 years are 0.59 times as likely to be healthy versus concurrent stunting and anaemia, when compared with children aged 0.5–1 years. However, Atsu et al. [15] found no significant effects for children aged 24–35, 36–47, and 48–59 months of being overweight and concurrently stunted, just as Adedokun [14] reported no significant effects for children aged 1–2 years compared to children aged less than 1 year.

#### 3.4.2. Child’s Sex

Another critical child-related characteristic reported in most of the studies under review is the sex of the child (See Table 3). Duah et al. [16] found that male children were more likely to have comorbid anaemia and diarrhoea compared to female children. In addition, Geda et al. [18] also reported a protective effect for female children compared with male children. Similarly, Tran et al. [19] found that male children are 22% less likely to be healthy relative to contracting concurrent stunting and anaemia compared to female children. Two studies (33%) reported no significant effects of a child’s sex on concurrent overweight and stunting [15] and diarrheal with an acute respiratory infection (ARI) [17].

Other child-related predictors extracted from the papers under review include child’s size at birth, child’s birth order and history of having fever within the two weeks before the survey. Adedokun [14] found a protective effect of multimorbidity of pneumonia, diarrhoea, and malaria fever for children whose mothers perceived they have ‘average size’ at birth relative to children whose mothers perceived they were ‘large size’ at birth, but found no significant effect for children born to mothers who perceived the children were born ‘small’ relative to children born ‘large’. Moreover, the paper found no significant effect on children’s birth order. There were more than four-folds harmful effects for children who had fever two weeks before the survey to contracting comorbid diarrhoea and anaemia when compared with children who had no fever.

### 3.5. Parental- and Household-Related Characteristics

Among the parental and household variables extracted from the studies selected for review, parental educational status had mixed conclusions. For instance, Mulatya & Mutuku [17] reported higher odds of having comorbid diarrheal and acute respiratory infection (ARI) among children of caregivers who had incomplete primary education relative to those without formal education. On the contrary, two other studies [16,18] found protective effects of fathers having secondary education and above relative to no education [16,18], and mothers had secondary education and above [18]. Geda et al. [18] found no significant effects on mothers and fathers who had no primary education. Contrary to expectation, Adedokun [14] found that increased maternal education status serves as a harmful effect for children who are cohabiting with ‘two or more’ childhood diseases versus ‘one or none’ of the diseases compared with children of mothers with no formal education. Atsu et al. found no significant effects on maternal education status [15]. Furthermore, household wealth status was reported in all the papers included in the study. Tran et al. found that the higher the household wealth quintiles, the more protective the children are being healthy relative to contracting concurrent stunting and anaemia [19]. Thus, the RRR of 0.49 for poorest wealth quintile vs. richest wealth quintile implies that households in the poorest wealth quintile have 0.49 times the risk of being healthy compared to households in the richest wealth quintile (i.e., they are less likely to be healthy than richer households and, consequently, more likely to have comorbidities than richer households) [19]. Other protective effects include maternal exposure to mass media [14] and the child living in a household with a size greater than five members [16]. However, Atsu et al. [15] reported harmful effects for children from the fourth wealth quintile households relative to poorest households.

### 3.6. Community- and Area-Related Characteristics

Being from North-East and South-East of Nigeria were associated with higher odds of multimorbidity among children under 5 years [14] compared to North-Central. Living in a rural area or place of residence was associated with a higher risk or odds of multimorbidity and, thus, more protective of being healthy relative to concurrent stunting and anaemia when compared with those who dwell in urban areas [19]. One study that used multilevel analysis [18] found protective effects for children from a community with mean maternal education at the cluster level [18].

### 3.7. Risk Factors with Directions by Authors

#### 3.7.1. Significant Effects-Related Characteristics by Authors

Table 4 displays the risk factors extracted from each paper and the direction of the significance level. For instance, Adedokun [14] found significant harmful effects with children cohabiting with two or more of pneumonia, diarrheal, and malaria if they are from middle wealth households, North-East and South-East, and a protective effect for children whose mothers were exposed to media. Similarly, Geda et al. [18] found a reduced likelihood effect of mother’s education on children cohabiting with concurrent stunting and anaemia. Furthermore, Atsu et al. [15] found significant increased likelihood effects of children who had diarrhoea or had ever been vaccinated to cohabit with overweight and concurrent stunting. Yet, Duah et al. [16] found a protective effect for children living with anaemia and diarrheal whose fathers had secondary or higher education. On the contrary, Mulatya & Mukutu [17], studying the cooccurrence of diarrheal and ARI among children under five years, found a protective effect with higher household wealth quintile and children of a caregiver with an incomplete education.

#### 3.7.2. No Significant Effects-Related Characteristics by Authors

The review also identified those variables that were not significant predictors of multimorbidity among children under five years (See Table 4). For instance, Adedokun [14] reported no significant effects of poorer, middle, and richest households on children contracting multimorbidity of diarrhoea, pneumonia, and fever versus combined of ‘none and one of the diseases’, when compared with children from the poorest household wealth quintile. In addition, the study found that North-West or South-South region of residence, child’s age is 1–2 years, born tiny, birth order at all levels, delivered in a health facility, lives in a household with access to an improved source of drinking water, and cooking method were no significant risk factors of multimorbidity of diarrhoea, fever, and pneumonia among children under five years of age in Nigeria. Similarly, child’s age is 24–35, 36–47, or 48–59 months; sex; religion of household head; maternal education status; household wealth quintile is second, middle and richest; area of residence; child’s mosquito net utilisation; a child diagnosed with malaria using the rapid test; and child had cough were not significant predictors of overweight with concurrent stunting among Ghanian children under five years of age [16]. Furthermore, a number of children aged < 5 years in a household; or the child is from a household with wealth quintile is poorer, middle, most prosperous; access to an improved source of drinking water; improved primary floor material; locality of residence is rural; and region of residence were not found to significantly predict multimorbidity in children [15]. Furthermore, Geda et al. [18] did not find that child’s nutritional status, sex, residence, exclusive breastfeeding between 0 and 6 months, and combined morbidity from diarrheal and ARI, caregivers had primary education completed, and secondary and above were significant risk factors of multimorbidity of comorbid anaemia and stunting. Mother’s age attained primary education level, father also had attained primary education level, the child never breastfed, and the level diet diversity score were not significant predictors of concurrent anaemia and stunting.

### 3.8. Distribution of Common Risk Factors across the Studies

Table 5 shows the summary of the common predictors across all the studies reviewed. Though the papers included studied of different cooccurrence of health conditions, two of these studies (Geda et al. [18] and Tran et al. [19]) studied similar disease conditions (Concurrent stunting and anaemia) but applied different statistical techniques. Tran et al.’s [19] presentation of results was different from other studies in that they had ‘healthy children’ versus ‘concurrent stunting and anaemia’ as the baseline.

The findings in this review show that child’s age and household wealth status were the two most common predictors of all the disease combinations, irrespective of their diverse conditions. Additionally, child’s sex was a predictor of anaemia and diarrhoea, and concurrent stunting and anaemia and maternal educational status were a predictor of diarrhoeal and ARI, and concurrent stunting and anaemia. Furthermore, paternal education status significantly predicts anaemia and diarrhoeal, and concurrent stunting and anaemia.

## 4. Discussion

The scoping review was conducted by searching for evidence of studies to establish the prevalence and the determinants of multimorbidity of childhood diseases among children under five years in Sub-Saharan Africa. The search was done first on 19 September 2021, and updated on 19 March 2022, covering publications between 1 January 1990, and 19 March 2022. It included studies that had national coverage and used nationally representative surveys. The selected papers were such that they applied classical regression analysis to determine the risk factors associated with multimorbidity of childhood diseases. This restriction of statistical techniques used was necessary because this study was a precursor of a larger study that focuses on the frequentist classical regression method, and to inform parallel comparisons of findings among studies. Out of the 261 studies found, only 6 met the inclusion criteria and were reviewed. The sample size for the studies included ranged from 2343 to 193,065 children under five years of age. There were six distinct disease conditions (Pneumonia, diarrhoea, malaria, being overweight, stunting, and anaemia) analysed in the included studies. One of the studies had three concurrent diseases, while the other five studies had two current diseases as their multimorbidity outcome of interest. The prevalence of the multimorbidity in these six studies ranged from 1.2% to 24.8%. The disease structure considered in the studies spans the most common childhood diseases in Low- and Middle-Income Countries (LMICs). It includes anaemia, diarrhoea, malaria, pneumonia, and nutritional deficiency indicators (stunting, underweight, wasting, overweight). Though it is difficult to draw a blanket comparison for these studies in view of divergent health status, the prevalence of cooccurrence of childhood diseases is fast becoming a public health burden in LMICs.

Adedokun (2020) [14] conducted the determinants of overlap among three outcome variables, which resulted in classifying multimorbidity as a count of four ordinal groups of ‘no disease’, ‘one disease only’, ‘two diseases’, and ‘three or more diseases’. The study used the 2013 Nigeria Demographic and Health Survey (2013 NDHS) data set, applying a generalised ordinal logistic regression model. Moreover, in Atsu et al.’s [15] study, the outcome variables of interest, stunting, overweight, and their concurrency (having overweight and stunted simultaneously) variables were classified into being overweight and concurrently stunted or not. The predictors were classified into three groups (hierarchical levels): distal, proximal, and intermediate. Finally, the prevalence ratios of the outcomes were computed. Though three models were formulated, they were not subjected to model fit to ascertain the model of best fit.

In the same way, Dual et al. [16] in their study, presented a multivariate analysis of determinants of comorbid anaemia and diarrhoea status. However, Mulatya & Mutuku [17] reported the overlap in the determinants of comorbid diarrhoeal and ARI. The children without multimorbidity and those with only one of the diseases were combined with no comorbidity. Furthermore, Geda et al. [18] had two separate analyses: (1) a count of the number of composite indexes of anthropometric failure (CIAF) of nutrition indicators a child has, (2) having concurrent stunting and anaemia or not. The predictors of concurrent stunting and anaemia were reported in this review. Similarly, Tran et al. [19] examined the comorbidity of anaemia and stunting, and they were compared with those of healthy children. Data from 43 Low- and Middle-Income Countries (LMICs) were pooled together to analyse three models: anaemia compared with comorbidity, stunting compared with comorbidity, and healthy compared with comorbidity children. The model of comorbidity and healthy children was reported with comorbidity as baseline.

Child’s age and household wealth quintiles are the two most important predictors of multimorbidity in under five children in SSA. Irrespective of the disease combination, they stood out as predictors in all the studies. As for the child’s age, there were no clear patterns in the direction of the association (with increasing age being associated with both in increased and decreased prevalence or multimorbidity), which may be partly because of the diversities in the diseases and study settings. Similarly, for household wealth status, findings showed that the higher the wealth status, the less likely the children will cohabit with multiple diseases in SSA. However, Tran et al. [19] found the similar conclusion from a pooled data set in 43 LMICs; the poorer the household, the more likely the children will cohabit with stunting and anaemia versus healthy children when compared with those from richest household. The possible reason for this is that wealthier households may have more resources to acquire those things that will impact good healthy living compared with those from poorer households. The study also found that female children are less likely to cohabit with multiple diseases compared with their male counterparts. There are assertions that the constitution of breast milk is a function of which gender is breastfeeding. Tran et al. states that girls breast feed longer than boys [19].

Parental education status was also an important predictor of multimorbidity. The more educated the parents are, the less likely the children will live with two or more diseases. Compared to children who live in urban areas, children who live in the rural are more likely to experience comorbidity [19]. In SSA countries, residing in a rural location always has disadvantages in terms of living standards, economic standing, and accessibility to healthcare. To avoid this co-morbidity, kids who live in rural regions require greater assistance.

### 4.1. Some Identified Study Gaps

Only in recent times, perhaps in the 2018 Nigeria Demographic and Health Survey (2018 NDHS) and subsequently, there was an expanded collection of more childhood diseases, including anaemia, acute respiratory infection (ARI), malaria, malnutrition, diarrhoea, fever, and sickle cell anaemia. Other than sickle cell anaemia, which is an inherited condition [20], information on anaemia, malaria, and malnutrition was objectively gathered using World Health Organisation (WHO) accepted practises. For diarrhoea, ARI, and fever, mothers were asked whether their kids had any of the illnesses in the two weeks prior to the survey. The accuracy of the information for these variables is determined by the ability of the mother to recall the accurate diagnosis of the illness. In a society like most SSA, where maternal literacy is low, it may not be possible to get accurate results. To the best of our knowledge, the current research demonstrates that no prior study has integrated such paediatric disorders with data objectively gathered from the survey as a multimorbidity framework. Studies to investigate the determinants of the cooccurrence of these three childhood diseases (anaemia, malaria, and malnutrition) and perhaps the six conditions (anaemia, malaria, and malnutrition, diarrhoea, ARI, and fever) are urgently needed to repose confidence in the outcome of the analyses obtained from these studies. In addition, almost all the included data sets were from nationally representative surveys with clear evidence of hierarchy. However, most of the existing studies did not take into consideration the multilevel structure and apply the appropriate statistical methods. Furthermore, most of the studies reviewed considered concurrency of two childhood diseases, and, therefore, classified the outcome of interest as a dichotomous variable and used the multivariate logistic or poison regression method, where children with concurrent of the two diseases were compared with children without the two diseases (none of the disease, and any one of the diseases).

### 4.2. Strength and Limitations

The strength of this review is that, to the best of the researchers’ knowledge, this is the first scoping review on risk factors associated with multimorbidity of childhood diseases that have used classical regression modelling techniques among children aged under five years in SSA countries with nationally representative survey samples. However, there are several limitations to this review. In view of the search techniques used, (i) it’s possible that some potential research papers were disregarded, especially the absence of truncation term in ‘risk factor*’; (ii) some papers that may have been written in languages other than English, but were associated with SSA countries, could have been lost; and (iii) the studies that were examined only used classical statistical regression techniques as their analytic tools. The use of Bayesian statistical methods in the analysis of possible publications was therefore disregarded. Finally, no evaluation of the potential risk of publication bias was done.

## 5. Conclusions

This scoping review sought to identify and describe studies that examined the relationship between socioeconomic, demographic, and environmental factors and the prevalence of multimorbidity among children under the age of five in Sub-Saharan Africa (SSA). Multimorbidity, until recently, has been associated with the adult population, but this has changed, especially in the LMICs, where more and more children are found to suffer from these emergent disease clusters. This scoping review on childhood multimorbidity is the first of its kind in SSA. The relatively few studies in this area of research showed the need for more studies to be done that will help policymakers to make sound policies to combat the growing trend of multimorbidity in LMIC and, thereby, improve the public health sectors. Particularly, the areas of intense need are that policymakers bear in mind that children from lower-income families appear to need more extensive and intense interventions in nations with considerable inequality, especially if resources are typically scarce. The findings in this study have also shown considerable clustering of childhood diseases around nutrition-related diseases in SSA. This requires more studies to be carried out that will inform integrated care-frameworks to be developed by clinicians, healthcare workers, and providers to reduce the burden of multimorbidity in SSA.

## Figures and Tables

**Figure 1 ijerph-20-01377-f001:**
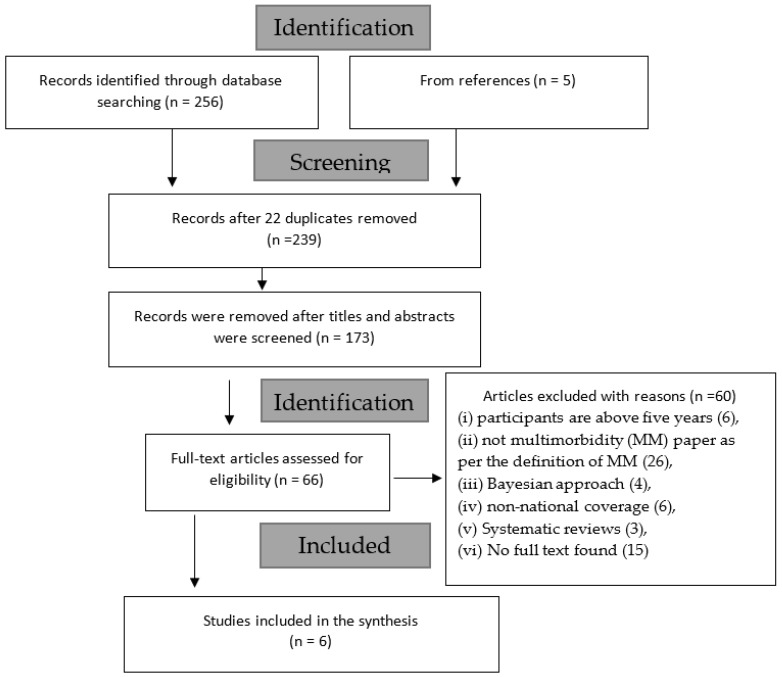
PRISMA Flow chart.

**Table 1 ijerph-20-01377-t001:** Draft Search strategy and terms for MEDLINE (Ovid).

S/N	Terms and Keywords	Results
1	Sub-Saharan Africa OR SSA OR low-and middle-income countries	37,452
2	Socioeconomic OR demographic OR contextual OR environmental OR community OR determinants OR risk factor OR predictor OR Association	3,298,141
3	comorbidity OR comorbidity OR multimorbidity OR multimorbidity OR multiple chronic conditions OR multi-diseases	451,419
4	Logistic regression OR multilevel regression OR multinomial logistic OR random-effects OR hierarchical OR fixed effects OR mixed-effects	55,708
5	1 AND 2 AND 3 AND 4	12
6	Limit 6 to human and English language and infant <to one year> OR preschool child <1 to 6 years>	
	Limit to last 30 years (1990 to 2020)	12

**Table 2 ijerph-20-01377-t002:** Distribution of the study characteristics.

Authors & Dates	Title	Objectives	Outcome Variables (Prevalence)	Sample Size (Participant’s Age)	Methods of Analysis	Country/Survey
Adedokun 2020 [14]	Correlates of childhood morbidity in Nigeria: Evidence from ordinal analysis of cross-sectional data	Correlates of childhood morbidity in Nigeria: Evidence from ordinal analysis of cross-sectional data	Diarrhoea, malaria and Pneumonia (9.0%)	27,571 (Under-5)	Generalised ordinal logistic regression mode	Nigeria/ Nigeria Demographic and Health Survey (NDHS 2013)
Atsu et al. 2017 [15]	Determinants of overweight with concurrent stunting among Ghanaian children	This paper presents the burden, the individual-level, and contextual determinants of overweight with concurrent stunting among Ghanaian children.	overweight with concurrent stunting (1.2%)	7550 (0–5 years)	A multivariable Poison regression model	Ghana/ Multiple Indicator Cluster Survey (MICS 2011)
Duah et al. 2020 [16]	Comorbid patterns of anaemia and diarrhoea among children aged under five years in Ghana: a multivariate complex sample logistic regression analysis and spatial mapping visualisation	To investigate the prevalence and independent predictors of comorbid patterns of anaemia and diarrhoea in children aged < 5 years in Ghana.	Anaemia and Diarrhoea (9.28%)	2343 (under-5 years)	multivariate logistic regression	Ghana/ Ghana Demographic and Health Survey (GDHS 2014).
Mulatya & Mutuku 2014 [17]	Assessing Comorbidity of Diarrheal and Acute Respiratory Infections in Children Under 5 Years: Evidence from Kenya’s Demographic Health Survey 2014. J. Prim. Care Community Health	This study seeks to assess the prevalence of comorbidity of pneumonia and diarrheal in children under-5 years, and to identify risk factors associated with comorbidity of pneumonia and diarrheal in children	Acute respiratory infection (ARI) and Diarrhoea (2.2%)	18,702 (under-5 years)	Multivariate logistic regression	Kenya/ Kenya Demographic Health Survey (KDHS 2014)
Geda et al. 2021 [18]	Multiple anthropometric and nutritional deficiencies in young children in Ethiopia: a multilevel analysis based on a nationally representative data	To examine the risk factors of cooccurrence of undernutrition and anaemia among children of age 6–59 months in Ethiopia based on nationally representative data	Anaemia and concurrent stunting (24.8%)	9218 (6–59 months)	Mixed effect logistic regression	Ethiopia/ Ethiopian Demographic and Health Survey (EDHS 2016)
Tran et al. 2019 [19]	Comorbid anaemia and stunting among children of preschool age in low- and middle-income countries: a syndemic	To determine the prevalence of comorbidity of anaemia and stunting, among children aged 6–59 months in low- and middle-income countries	Anaemia and concurrent stunting (21.5%)	193,065 (6–59 months)	Multinomial logistic models	Multi-countries Demographic and Health Surveys (DHS 2005–2015)

**Table 3 ijerph-20-01377-t003:** Distribution of the extracted risk factors of multimorbidity.

Child-Related Variables	
Child’s age	Protective effects <1 year (ref); 3 years and above, aOR = 0.43 (0.34–0.55) [14]. 0–11 months (ref), 12–23 months, aPR = 0.991 (0.982–0.999) [15]. 0.5–1 year (ref), 1–2 years, aRRR = 0.59 (0.55–0.64), 2–3 years, aRRR = 0.87 (0.80–0.94) [19] ^‡^. Harmful effects 24–59 months (ref), 6–23 months, OR = 2.17 (1.42 to 3.33) [16]. <6 months (ref), 6–11 months, aOR = 3.48 (2.02–5.99), 24–35 months, aOR = 2.84 (1.71–4.70) [17]. 0–23 months (ref), 24–35 months, aOR = 6.55 (5.26–8.15), 36–59 months, aOR = 4.29 (3.43–5.36) [18]. 0.5–1 year (ref), 3–4 years, aRRR = 1.27 (1.18–1.37). 4–5 years, aRRR = 1.86 (1.72–2.01) [19].
Child’s sex	Protective effects Male (ref), Female, aOR = 0.84 (0.74–0.93) [18]. Female (ref), Male, aRRR = 0.78 (0.75–0.81) [19]. Harmful effects Female (ref), Male, OR = 1.50 (1.04 to 2.16) [16].
Child’s birth size	Protective effects Large (ref) Average size at birth, aOR = 0.68 (0.57–0.82) [14].
Diarrheal status	Harmful effects No (ref), Had diarrhoea (Yes), aPR = 1.019 (1.006–1.032) [15].
Fever status	Harmful effects No (ref), Had fever (Yes), OR = 4.37 (2.94 to 6.50) [16].
Vaccination status	Protective effects No (ref), Ever been vaccinated, aPR = 0.997 (0.960–0.995) [15].
Breastfeeding status	Protective effects No (ref), Ever been breastfed, aPR = 0.995 (0.984–1.006) [15].
Parental-related variables	
Maternal education status	Protective effects No education (ref), Secondary education and above, aOR = 0.64 (0.48–0.86) [18]. Secondary or higher (ref), Primary education, aRRR = 0.43 (0.41–0.46) No formal education, aRRR = 0.20 (0.19–0.21) [19]. Harmful effects No education (ref), Incomplete primary education, aOR = 1.66 (1.11–2.50) [17]. Primary education, aOR = 1.29 (1.13–1.46), secondary/higher, aOR = 1.42 (1.23–1.65) [14]
Paternal education status	Protective effects No formal education (ref), Secondary or higher education, OR = 0.57 (0.33 to 0.97) [16]. No education (ref), Secondary or higher education, aOR = 0.81 (0.65–1.00) [18].
Caregiver’s age	Protective effects 15–19 years (ref), 30–34 years, aOR = 0.49 (0.28–0.85), 40–44 years, aOR = 0.47 (0.23–0.95) [17].
Maternal exposure to media	Protective effects Never exposed (ref), Exposed to media, aOR = 0.82 (0.67–0.99) [14].
Household-related variables	
Wealth status	Protective effects Poorest (ref), Richer wealth households, aOR = 0.83 (0.70–0.99) [14]. Poorest (ref), Richer household, OR = 0.38 (0.16 to 0.89) [16]. Poorest (ref), Middle wealth quintile, aOR = 0.58 (0.39–0.85), Highest wealth quintile aOR = 0.43 (0.24–0.77) [17]. Poorer/poorest (ref), Middle wealth quintile, aOR = 0.73 (0.61–0.87) Richer/richest, aOR = 0.64 (0.54–0.75) [18]. Richest (ref), 4th wealth quintile, (Richer) aRRR = 0.71 (0.65–0.77), Middle wealth quintile, aRRR = 0.62 (0.57–0.68), 2nd quintile (poorer), aRRR = 0.55 (0.50–0.60), Poorest wealth quintile, aRRR = 0.49 (0.45–0.53) [19]. Harmful effects Poorest (ref), 4th wealth quintile, aPR = 1.011 (1.001–1.021) [15].
Number of under-5 years	Harmful effects 0–1 (ref), Had two children aged < 5 years, OR = 1.80 (1.14 to 2.84) [16].
Household size	Protective effects 1–5 (ref), Had ≥ 6 members, OR = 0.46 (0.28 to 0.75) [16].
Ethnicity of household head	Harmful effects Akan (ref), Head is of the Ewe tribes, aPR = 1.023 (1.000–1.046) [15]
The religion of household head	Orthodox (ref), Religion (others, besides being Orthodox) aOR = 1.37 (1.17–1.61) [18].
Sanitation	Median (ref) Sanitation score aOR = 1.12 (1.01–1.24) [18].
Community-related variables	
Maternal education status	Protective effects Mean (ref), maternal education at cluster level, aOR = 0.94 (0.90–0.98) [18].
State-related variables	
Region of residence	Harmful effects North-Central (ref), North-East, aOR = 5.34 (3.86–7.39) South-East, aOR = 3.17 (2.15–4.66) [14].
Place of residence	Protective effects Urban (ref), Rural, aRRR = 0.72 (0.67–0.77) [19].

aOR = adjusted odd ratios, aRRR = adjusted relative risk ratios, aPR = adjusted poison ratio, ^‡^ Tran et al. [19] reported the aRRR for being healthy relative to concurrent stunting and anaemia as baseline.

**Table 4 ijerph-20-01377-t004:** Summary of risk factors with directions by authors.

Authors and Date	Harmful Effects (Increased Likelihood)	Protective Effects (Decreased Likelihood)	No Significant Effects
Adedokun. 2020 [14]	Child is from middle wealth households; Child’s region of residence is: North-East and South-East.	Child is 3 years and above, Mothers are exposed to media, Child’s average size at birth.	Mother’s age difference, Maternal education, being in the poorer, richer, and richest household, North-West, South-South, child’s age is 1–2 years, child is born small, Birth order, delivered in health facility, improved source of drinking water and cooking method.
Atsu et al. 2017 [15]	Ethnicity of household head is of the Ewe tribes, fourth wealth quintile, child is with diarrhoea, child ever been vaccinated.	Child’s age is 12–23 months, child ever been breastfed	Child’s age is 24–35, 36–47, 48–59 months; sex; religion of household head, maternal education status, wealth index is second, middle and richest, area of residence, mosquito net utilization malaria rapid test, child had cough
Duah et al. 2021 [16]	Child’s age between 6 and 23 months, Child is male gender, History of fever, and living in a household with two children aged < 5 years.	Father having secondary or higher education, Living in a household with ≥6 members, and Living in a richer household.	Number of children aged < 5 years; household quintile is poorer, middle, richest; improved source of drinking water, main floor material, locality of residence is rural, region of residence.
Mulatya & Mutuku 2014 [17]	Child’s age is 6–11 months, and caregivers have incomplete primary education.	High wealth quintile, and older caregivers.	Nutritional status of a child, sex, residence, exclusive breastfeeding between 0 and 6 months, and combined morbidity from diarrheal and ARI, caregivers had Primary complete, and secondary and above
Geda et al. 2021 [18]	Child’s age is 24–35 months, 36–59 months, religion (others), sanitation score	Child’s sex is female, child’s mother has secondary education and above, child’s father has secondary education and above, wealth index is middle, and richer/richest, and mean maternal education at cluster level.	Mother’s age, mother has primary education level, father has primary education level, child never breastfed, and diet diversity score.
Tran et al. 2019 [19]	The child’s age is 3–4 years, and 4–5 years	Child’s sex is boy; child’s age is 1–2 years, 2–3 years, place of residence is rural; child’s mother has primary education, has no formal education; household wealth index is 4th quintile (Richer), middle, 2nd quintile (poorer), and poorest.	

**Table 5 ijerph-20-01377-t005:** Distribution of common risk factors across the studies.

Authors	Adedokun 2020 [14]	Atsu et al. 2017 [15]	Duah et al. 2020 [16]	Mulatya & Mutuku 2014 [17]	Geda et al. 2021 [18]	Tran et al. 2019 [19]	
Conditions studied	Pneumonia, diarrhoea, and malaria	overweight with concurrent stunting	Anaemia and Diarrhoea	Diarrheal and acute respiratory infection (ARI)	Concurrent stunting & anaemia	Concurrent stunting and anaemia	
Child-related variables							
Child’s age	Significant	Significant	Significant	Significant	Significant	Significant	6
Child’s sex			Significant		Significant	Significant	3
Child’s birth size	Significant						1
Diarrheal status		Significant					1
Fever status			Significant				1
Vaccination status		Significant					1
Breastfeeding status		Significant					1
Parental-related variables							
Maternal education status				Significant	Significant	Significant	3
Paternal education status			Significant		Significant		2
Caregiver’s age				Significant			1
Maternal exposure to media	Significant						1
Household-related variables							
Wealth status	Significant	Significant	Significant	Significant	Significant	Significant	6
Number of under-5 years			Significant				1
Household size			Significant				1
Ethnicity of household head		Significant					1
The religion of household head					Significant		1
Sanitation					Significant		1
Community-related variables							
Maternal education status					Significant		1
State-related variables							
Region of residence	Significant						1
Place of residence						Significant	1

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
