# Peer review of "Risk Factors Associated with Multimorbidity among Children Aged Under-Five Years in Sub-Saharan African Countries: A Scoping Review"

_ijerph, 2023, doi:10.3390/ijerph20021377_

Round 1

Reviewer 1 Report

Dear authors,

the manuscript aims to identify risk factors for multimorbidity in children under 5 in SSA by conducting a scoping review.

The identified gaps are important for researchers and politicians and an important sign to gather more information on this topic.

However, before publishing the manuscript needs to be revised.

There are some minor but also some major revision points I need to mention.

First, I am not convinced by the discussion/conclusion. You conclude that there are gaps that need to be filled which is correct. However, in your manuscript, you also describe the risk factors for multimorbidity but do not explain these results in the discussion or conclusion. From what I understood from the results, the effect of the risk factors on multimorbidity depends on which diseases were summarized in the multimorbidity. This makes sense when the diseases are quite different. Although it might also be possible for a risk factor to influence diseases in similar ways. I would like to see an analysis of the data on this and a description of what this means in the discussion. The discussion is missing an interpretation of the results besides the identification of gaps, especially as your study aim was to evaluate the relationship between socioeconomic, demographic, and contextual factors and the prevalence of multimorbidity in the found papers.

Second, the documentation of the search strategy is incomplete. You mention keywords and how they were connected but not if you used mesh terms or only searched in title/abstract or full text. Further, I was surprised to see that you didn’t use truncation e.g. risk factor*. Please provide more information.

Third, why did you only include papers with regressions? If there’s a reason, it would be good to include it in the manuscript.

Fourth, Table 3. There are differences in risk factors being a protective or harmful effect depending on the used statistical measurement e.g. aOR or aRRR. You describe this a little bit in the text but it is very confusing in the table. The differences occur because of the different effects some risk factors have on different diseases. Table 3 shows that it is difficult to summarize the results. As I said above, the interpretation of the data is important as well. Table 3 does not allow the reader to understand your results without your interpretation. A table has to be self-explanatory, so I suggest rearranging and adapting table 3. 

Some additional points:

-          In the methods, you mention that the search was done on 19 March 2022. In the discussion (line 271) you write that the search was done between 19 September 2021 and 19 March 2022. Please clarify when the search was done and what is correct.

-          Did I understand correctly that the title & abstract screening was done twice by the same person? If yes, why? Please provide more information on how the screening and data extraction process was done.

-          Why didn’t you register a protocol for your review?

-          Why didn’t you analyze the publication bias?

-          I would suggest including the exclusion reasons for full-text screening as well in the PRISMA flowchart. You only mention the reasons in the text.

-          You are using “developed”, “developing” and “underdeveloped” to describe the SSA countries. Especially “underdeveloped” can be seen as a stigmatizing word. I would avoid these phrases and use high-, middle- and low-income countries instead.

-          Table 3. Sometimes there are no references for some numbers. I am not sure if this means that it is the same reference as for the numbers in the line above or below. Please clarify and adapt the table.

-          Table 3. What is meant by “other religions”? See my point above about self-explanatory tables.

Some minor points:

-          When you cite, sometimes the dot is behind and sometimes before the brackets of the citation. Please check again which way you want to use.

o   .[..] vs [..].

-          Line 29: “of a huge knowledge”

-          Line 104: The dot is missing at the end of the sentence.

-          Line 238 and line 248: Check the spelling of the geographical Region. Dash or no dash.

-          Line 297: LMIC, the abbreviation is not explained.

All the best and Merry Christmas

Author Response

Please find in the attached file the authors' responses to reviewer 1's comments

Reviewer 2 Report

This paper tries to lay out risk factors associated with multimorbidity among children under 5 years in Africa by reviewing relevant journal articles which appeared in refereed international journals. This review paper should be highly appreciated in a situation where reducing the prevalence of multimorbidity in children aged under 5 years in developing countries, especially in sub-Saharan Africa, is an important policy issue to attain the SDGs goals. However, if possible, the following points need to be addressed.

1) It would be helpful for readers to explain what similarities or differences there are between the factors noted as being associated with the prevalence of a specific disease and those noted as being associated with the incidence of multimorbidity.

2) Some of the factors noted as being associated with the prevalence of multimorbidity may be interrelated. For example, the factor of regional differences may be related to economic disparities and the amount of maternal and child support programs provided by local governments among regions. It seems to me that authors should not only to simply discuss which factors are associated with the incidence of multimorbidity, but also to discuss the possibility that each factor is related to each other.

3) Careful editing is required to submit the revised version.

Author Response

Please find in the attached file the authors' responses to reviewer 2's comments

Round 2

Reviewer 2 Report

Thank you for revising the paper. I think that the paper was largely revised in line with comments. However, the following points, which may be very trivial from the grammatical viewpoint but could jeopardize the academic value of the paper, must be corrected.

In the sentence in line 207, the phrase "relative to" is duplicated. Either one should be deleted.

Although the authors used the word "et al" in the revised sentences and tables, it should be replaced with the word "et al.".

As for Table 5, it appears on page 12 in the revised version, but the title of the table is not shown.

I am not a native English speaker; however, as mentioned above, I am a bit skeptical to the accuracy of English expressions, especially in the revised sections. I am wondering whether all the authors had really read through the revised paper with a great care. Although the manuscript is sufficiently revised in terms of content, it would be necessary for all authors to go through the paper again and, if any, correct all grammatical errors.

Author Response

Thank you, for your valuable comments, which have helped to enhance the academic value of the paper.

Please find in the attached file the responses to your comments (Round 2)
